# A Comparative Study of the Relationship among Antecedents and Job Satisfaction in Taiwan and Mainland China: Employability as Mediator

**DOI:** 10.3390/ijerph16142613

**Published:** 2019-07-23

**Authors:** Michael Yao-Ping Peng, Chun-Chun Chen, Hsin-Yi Yen

**Affiliations:** 1School of Economics and Management, Xi’an University of Posts & Telecommunications, Xi’an 710121, China; 2School of Management, Beijing Union University, Beijing 100101, China; 3Department of International Business, Providence University, Taichung City 43301, Taiwan

**Keywords:** employability, job security, job satisfaction, organizational support

## Abstract

Previous studies of the relationship between job security and job satisfaction were mostly conducted on research samples in Asia from the perspective of oriental culture; however, under the same cultural background, different social systems might lead to different cognition outcomes. Therefore, this study examines the job security and organizational support of Taiwan and mainland China employees from the perspectives of competence enhancement motivation, and investigates the relationship between employability and job satisfaction. Adopting judgmental sampling, a total of 1307 valid questionnaires were collected from Taiwan and mainland China employees. The path relationship of the two groups was examined through structural equation modeling (SEM) by using analysis of moment structure (AMOS). Results show that job security and organizational support are positive for employability and job satisfaction. Employability has a positive influence on job satisfaction. Additionally, employability has a mediating effect of job security and organizational support on job satisfaction.

## 1. Introduction

In recent years, global governments and scholars have been paid great attention to the study of employability, covering the development of students’ cultivation for employment and the enhancement of employees’ ability. As De Vos, De Hauw and Van der Heijden [1] indicated, employability means the individual’s ability of acquiring knowledge, skills and other characteristics in order to meet the needs of employers and exert professional potential. The research results and contributions may diverge, as related issues of employability differ in specific research situations and research designs. Some studies have pointed out that related issues of employability can be divided into two categories. The first category is to examine the influence of organizations and individuals on employability from the employee’s perspective. The second category is to examine the influence of universities and individuals on employability from the student’s perspective. This study focuses on organizational behavior to discuss the direct correlation between the motivations and learning ability gained from education and training, and the overall economic performance of Western countries. Hence, the issue explored in this study emphasizes how employees in the labor market can quickly adapt to the changing work environment, job demands and emerging technologies [2]. In addition, as employability policies are almost made specific to the overall labor force rather than to the unemployed and underrepresented groups, related research relates to meeting job security rather than assisting the individual to get a job [3]. Therefore, one of the issues discussed in this study is job security.

Basic changes in the labor market have provided inspiration for this research, namely, how to reduce job insecurity among employees [4]. Therefore, for employees, employability comes with two interesting revelations. Firstly, employability is favorable to the career development of employees [5]. Secondly, the development of employability stems from more job insecurity [6]. This shows that the potential consequences of employability-related issues can be regarded as labor market assessment indicators, such as salary, employment experience, and non-voluntary mobility rate [7].

Most previous studies of organizational behaviors have concentrated on how to enhance employees’ job satisfaction through the psychological factors of leaders, the organization and employees, but few studies have examined the relevance of the relationship between the employability of employees and their life and job satisfaction [8]. De Vos et al. [1] discussed the career success factors of employees from the perspective of personal ability, namely, how employees exert their professional potential through acquiring knowledge, techniques, skills and other valuable characteristics [9,10,11]. In other words, only by enhancing the competitiveness of individuals in the job market and acquiring problem-solving skills, can they enhance their psychological satisfaction. In addition, few previous studies have combined the organization with individual initiatives to enhance the development of employees’ employability [8,12]. In this regard, in addition to the research framework of De Vos et al. [1], the relationship between employees’ employability and satisfaction was also studied from the perspective of job security.

On the other hand, scholars believe that interregional comparison results can provide significant insights, so that most scholars adopted interregional methods to compare the different research results in different contexts [13]. According to the World Happiness Report in 2016, the happiness indexes of Taiwan and mainland China were 34 (6.379) and 83 (524.5), respectively. However, in terms of the indexes of economic development, the GDP growth rate of mainland China was about 6.5%, while that of Taiwan was only 1.6%. Contrasting to the happiness index, the huge difference in GDP growth is worthy of further study. Therefore, in this study, employees in mainland China and Taiwan were taken as the research samples of the interregional comparison in order to learn about the relevance of the research variables.

To summarize, there are three contributions of this study. Firstly, from the perspective of capability development and employability, it provides some insights into the career success factors and addresses a vacancy of related research. Secondly, from the perspective of the employees, it discusses the growth patterns of employability based on organizational factors, and adds job security as the factor stimulating employees to enhance personal abilities and employability, thus enriching the theoretical framework of employability. Thirdly, from the perspective of the huge differences in GDP growth and happiness index, employees in Taiwan and mainland China were studied to examine the differences among the variables in order to provide more valuable practical significance.

## 2. Theory and Hypotheses

### 2.1. Employability 

In recent years, research related to employability has increasingly reflected concern of scholars. Scholars have studied the meaning of employability and the causality between employability and other factors through the design of research situations and methods, and the integration of theoretical and practical analysis [14,15]. Van der Heijde and Van der Heijden [11] defined employability as an individual’s appropriate application of competence and continuous acquisition and creation of essential work skills in order to accomplish all tasks, and to adapt to changes in the internal and external labor market [10,16,17]. Therefore, McQuaid and Lindsay [18] proposed that, in addition to the factor of basic education, factors like interpersonal relations, external factors and personal conditions that cannot be acquired in higher education should also be considered.

In addition, the Confederation of British Industry regarded employability as the technology quality and ability owned by individuals to meet the changing requirements of employers and customers. The Australian Chamber of Commerce and Industry in 2002 started from the perspective of enterprise development and noted that employability can not only help the individual to get a job, but also promote enterprise development, i.e., exerting personal potential is conducive to proposing the directions and strategies for success. Thus, it may be known that employability is a kind of social mental construct composed of subjective and objective aspects [1], namely, effectively responding to the requirements of the employment environment that can help individuals adjust their own characteristics, behaviors, cognitions and emotions, and thus maintain their adaptability and flexibility in the workplace [8]. In this regard, in this study, the employability scale developed by Pan and Lee [19] based on the research results of Andrews and Higson [20] was referred to and it was suggested that the employability could include the general ability for work, professional ability for work, attitude at work, and career planning and confidence.

### 2.2. Job Security

Maslow’s hierarchy of needs theory [21] holds that the human beings are inspired by five levels of demands, namely, physiology, security, safety, esteem and self-actualization. Among these, security demand is at the lower level, i.e., the body is not violated and the mentality is not threatened by fears. Maslow’s follow-up study [22] showed that, in addition to physical safety, employees in the organization also need job security in the forms of job stabilization and seniority security. With respect to job security, job insecurity can reflect employees’ psychological reaction to the job through subjective and objective indicators [23]. Sverke, Hellgren and Näswall [24] conducted integrated analysis on job insecurity and related research results, and found that the single indicator “Do you worry about losing your job in the future?” can effectively measure job security and predict employees’ job satisfaction, job performance, and their trust in the organization. In this study, based on the research results of Cuyper et al. [16], the definition of job insecurity was identified as employees’ perceptions about potential involuntary job loss [24].

Furthermore, based on the competence enhancement mode, this study was also made to infer the influence of employee’s job security on their employability. The inference process was contrary to the results of Berntson and Marklund [25], Cuyper et al. [16], Forrier and Sels [26] and Sverke et al. [24] who, based on appraisal theory, argued that employees will regard the current labor market and economic turmoil as a challenge rather than a threat; thus, high employability is negatively related to job insecurity. Nevertheless, some studies have shown that the psychological process and performance of individuals from non-Western societies in the pursuit of achievements are significantly different from the theoretical basis established in Western societies [13,27]. In other words, the hierarchy of needs and the cognition-oriented method has failed to fully explain the psychology and behavior models of non-Western individuals, especially those employees in the Asian region. For example, in the face of personal success or failure, American employees tend to attribute success to internal factors and attribute failure to external causes. On the contrary, this attribution model has not been proved in Asian employees, for they tend to attribute failure to internal factors, especially insufficient effort [28,29]. The samples of this study are selected from Asian countries. When employees are subject to job insecurity, they will attribute it to personal incompetence and further try to enhance their employability in order to reduce their sense of job insecurity. Therefore, a hypothesis of this study is as below.

**Hypothesis** **1** **(H1).**
*Employees’ job security is negatively related to their employability.*


In addition, job insecurity will do harm to employees’ physical and mental health and have a negative impact on job satisfaction [30,31,32,33], which further reduces employees’ organizational commitment [34], trust and sense of loyalty [35], and give rise to employees’ retreat and turnover intention [36]. Even so, among all the variables of working attitude and behavior, this study mainly focuses on two variables, namely, job satisfaction and organizational commitment.

Ashford, Lee, and Bobko [37] have argued that job insecurity is a constellation of perceptions which have great influence on employees’ job satisfaction and organizational commitment. Many studies have verified that employees’ job insecurity has a negative impact on their job satisfaction [24,30,31,32,33,38]. However, from the perspective of the sense of job security, when employees have high job security, they will show high job satisfaction [39,40], and are influential, to a certain extent, on job satisfaction no matter the level of security or insecurity. Therefore, a hypothesis of this study is as below.

**Hypothesis** **2** **(H2).**
*Employees’ job security is positively related to their job satisfaction.*


### 2.3. Perceived Organizational Support

Perceived organizational support (POS) refers to employees’ psychological perception in assessing the organization’s attention to employees’ contributions and welfare. Previous studies of organizational support are mostly based on the principle of reciprocity proposed in organizational citizenship behavior theory and social exchange theory, namely, employees will work hard and offer high loyalty as the return to the tangible benefits given by the organization when they experience high organizational support. Empirical studies have shown that employees’ perceived organizational support will enhance their working attitude, such as their responsibilities to the organization [41,42], the adaptation to the environment [43], organization-based self-esteem and trust in the organization [44]. De Vos et al. [12] also have pointed out that perceived organizational support is positively related to employees’ career development and job satisfaction. Therefore, a hypothesis of this study is as below.

**Hypothesis** **3** **(H3).**
*Employees’ perceived organizational support is positively related to their job satisfaction.*


Changes in the labor market environment further diversify the requirements for employees in terms of work skills and abilities, and require them to timely adjust and acquire new expertise and skills to meet job requirements. However, learning new expertise and skills requires heavy investments of time and resources. If employees fail to get real-time access to the expertise to solve the problems, their work pressure will be increased and positive perceptions, such as self-confidence and sense of satisfaction, will be decreased. That is to say, employees’ sense of crisis caused by poor performance will give rise to their job insecurity. Relatively, Schneider, Brief and Guzzo [45] have found that organizational support for employees’ capability development can not only enhance their professional knowledge and skills, but also can enhance their employability. De Vos et al. [1] have pointed out that in addition to offering formal or informal learning activities to develop employees’ skills, the organizational culture will also strongly support employees to learn new knowledge and skills, and thus positively influence their employability. Similarly, employees’ high perceived organizational support suggests that they think the organization attaches great importance to their contributions, so they will maintain a positive mood to cope with pressure and enhance their organizational citizenship behavior, and hence enhance their job security. Therefore, hypotheses of this study are as below.

**Hypothesis** **4** **(H4).**
*Employees’ perceived organizational support is positively related to their employability.*


**Hypothesis** **5** **(H5).**
*Employees’ perceived organizational support is positively related to their job security.*


### 2.4. Job Satisfaction

Fred [46] has argued that job satisfaction is related to employees’ emotional perception and reaction to the job contents, the working environment, and the gap between the actual remuneration and the expected remuneration [47]. This includes intrinsic satisfaction [48], such as sense of achievement, self-esteem, independence, feedback and control power, and extrinsic satisfaction, such as supervisors’ recognition, harmonious relationships among colleagues, favorable working environment, and good welfare, salary and promotion prospects [49,50]. Wernimont [51] pointed out that job satisfaction will ensure employees voluntarily remain cooperative with the organization to achieve task goals, maintain high loyalty to the supervisors, stand together with the organization to weather the storm and take pride in being a part of the organization. Seashore and Taber [52] have argued that job satisfaction can be regarded as an early warning indicator of internal conflicts and the important basis for the organization to formulate relevant management policies [50]. Therefore, in this study, job satisfaction is regarded as the employees’ perception and emotional evaluation of the current job, which includes intrinsic satisfaction and extrinsic satisfaction.

Although job satisfaction can be regarded as a predictor of job performance, and a very recent study related employability with life satisfaction and quality of life [8], few empirical studies have explored the relationship between employees’ employability and job satisfaction [1,53]. De Vos et al. [1] have explained the problem from the perspective of human capital theory. They have emphasized that investments in individuals’ skill development will increase their value in the job market and make human capital elements, such as the acquisition of technologies and skills, positively influence job satisfaction. The study interprets employees’ employability from the perspective of problem-based learning, i.e., employees’ problem-solving and learning process will enhance their expertise and skills, which will indirectly improve their employability. Furthermore, after problems are solved, employees will gain positive self-confidence and a sense of achievement, indicating that the need of self-actualization proposed by Maslow’s hierarchy of needs theory can be satisfied. Therefore, a hypothesis of this study is as below.

**Hypothesis** **6** **(H6).**
*Employees’ employability is positively related to their job satisfaction.*


### 2.5. Differences in Interregional Contexts

According to the theory of national culture, enduring national values that shape organizational behavior differ between nationalities [54]. Nationality reflects the shared cultural system of members of a certain country [55]. As such, organizations within a given country share common characteristics and features, such as norms, values and social system, that derive from their shared experiences [56]. Differences in nationality are likely to influence perceptions of the workplace based on how employees (i) appraise and (ii) respond to working conditions [54]. For example, differences in normative prescriptions, which stem from nationality, may shape primary appraisals of what constitutes demanding conditions at work [57]. Taiwan is a democratic social system in which welfare is highly advocated. In Taiwan, it is believed that only by the redistribution of resources can the society actually realize social justice. Similarly, Taiwan enterprises also advocate giving employees equal social rights and trying to materialize welfare assistance as a guarantee for equal rights. By contrast, mainland China is a communist country and, for a long time, has been implementing a planned economic system. Therefore, the systemic environment of highly-concentrated administrative command and soft constraints will give rise to corresponding management methods and management personnel.

Raymond, Hollenbeck, Gerhart and Wright [58] have pointed out that the human resources management of mainland enterprises is not only in the pursuit of the enterprises’ profit growth and long-term development, but also of the promotion of personal official careers. Some Taiwan manufacturers set up factories in mainland China. The different social systems of both sides of the Taiwan Strait have resulted in differences in organizational support, job security, employability and job satisfaction. Whether these differences will affect Taiwan manufacturers in the effective use of human resource management strategies needs to be further explored. Therefore, a hypothesis of this study is as below. 

**Hypothesis** **7** **(H7).**
*There are differences between the samples in Taiwan and those in mainland China.*


In addition, the research framework is constituted according to the research purposes and hypotheses, as shown in Figure 1.

## 3. Methodology

### 3.1. Participants and Procedures

This aim of this study is to discuss the causality of employees’ employability. Questionnaires were delivered to the service industries to explore the interaction between the organization and employees, and the differences in the employee development under the different social backgrounds of Taiwan and mainland China. All the surveyed were informed by their company leaders who approved of employees in participating in this research program. In addition, the researchers explained the contents of the questionnaires to the surveyed in advance to eliminate any possible misunderstandings. All of the surveyed participated voluntarily.

In total, 1000 copies of the questionnaire were delivered to front-line employees in Taiwan service industries, including enterprises from the catering industry, information service industry and the tourism industry. A total of 769 completed questionnaires were collected and 759 copies found to be valid. Hence, the collection rate was 75.9%. In total, 1000 copies of the questionnaire were also delivered to employees of mainland China service industries, and most questionnaires were collected from Fujian and Zhejiang provinces. A total of 557 completed questionnaires were collected and 548 copies found to be valid. Hence, the collection rate was 54.8%. The background variables of the samples are shown in Table 1.

### 3.2. Measures

Employability is a social psychological construct which includes subjective and objective aspects [1]. It can help employees to adjust their own characteristics, behaviors, cognition and emotion according to the demands of the employment environment, so as to maintain their adaptability and flexibility in the workplace. In order to understand employability, this study adopted the variable measurement of Pan and Lee [19], which includes “General ability for work” (8 questions), “Professional ability for work” (4 questions), “Attitude at work” (3 questions) and “Career planning and confidence” (3 questions). 

Job insecurity refers to employees’ perceptions about potential involuntary job loss. Job insecurity was assessed using five items developed by Francis and Barling [59] for the current study.

Perceived organizational support (POS) refers to employees’ belief from a psychological development perspective to assess the organization’s attention to employees’ contributions and welfare. Perceived organizational support for competency development was also measured by the scale developed by De Vos et al. [1], based on the same qualitative case study referred to earlier. Twelve items were selected that assess the extent to which respondents experience support of competency development from the organization.

In this study, job satisfaction is defined as employees’ satisfaction with the working environment, which is their general attitude towards the job. This scale is developed by Janssen [60], which includes 7 questions. 

Respondents had to indicate to what extent they agreed with these statements with a five-point Likert scale (1 = totally disagree; 5 = totally agree). All scales are shown in Table 2.

### 3.3. Analytical Strategy

We tested the hypothesized model and included paths via structural equation modeling. For construction with a higher-order factor structure (employability), we reduced the number of parameters to be estimated following the partial aggregation method [61,62]. This procedure involves averaging the responses of sub-sets of items measuring a construct. Because job security, job satisfaction and organizational support were uni-dimensional constructs, we followed the procedure recommended by Little et al. [62] to create two parcels of randomly selected items to serve as indicators for these variables.

## 4. Analysis and Results

### 4.1. Reliability and Validity

All scales used in this study were found to be reliable, with Cronbach’s α ranging from 0.83 to 0.96. Table 3 shows the reliability of each scale, and the factor loadings for each item therein. In order to gauge validity, this study employed confirmatory factor analysis (CFA) using Linear Structural Relations (LISREL) 8.54 to verify the construct validity (both convergent and discriminant) of the scales. The measurement model provided a good fit to the data: χ^2^ (378) = 1271.404; Tucker–Lewis Index (TLI) = 0.95; comparative fit index (CFI) = 0.96; standardized root mean square residual (SRMR) = 0.023; root mean square error of approximation (RMSEA) = 0.051. Hair, Black, Babin, Anderson, and Tatham [63] recommended convergent validity criteria as follows: standardized factor loading of higher than 0.7; average variance extracted (AVE) ranges between 0.416 and 0.782; and composite reliability (CR) ranges between 0.703 and 0.934. The evaluation standard for discriminant validity is the square root of AVE for one dimension greater than the correlation coefficient with any other dimension(s). As Table 3 indicates, all three criteria for convergent validity were met, and correlation coefficients were all less than the square root of the AVE within one dimension, suggesting that each dimension in this study had good discriminant validity.

### 4.2. Multi-Group Testing

It has been confirmed that the measurement pattern is stable. However, in order to avoid the data-driven pattern and theory from being overgeneralized, the suggestions of Hair, Black, Babin, Anderson and Tatham [63] were taken to divide the sample data into two groups randomly (344 and 275, respectively). Moreover, multi-group testing was combined with bootstrapping to gradually control the pattern parameters of the groups, including Unconstrained, Measurement weights, Structural weights, Structural covariance, Structural residuals and Measurement residuals. The nested models were developed from different limitations of χ^2^ difference quantity to undertake significance analysis, in order to determine the reasonability of those parameters in controlling the two groups. The results are shown in Table 4. 

Analysis results show that the value of each pattern mode of *χ*^2^/*df* ranges from 2 to 3, the RMSEA ranges between 0.54 and 0.56, and ECVI is in 90% of the confidence interval. It can be learned from Table 4 that the values of the weighted measurement model, weighted structure model, covariance structure model and residual structure model *χ*^2^ reached a significant level. This shows that the models have good between-group invariances. In addition, the normed-fit index (NFI) added value of each model is less than 0.05, which complies with the recommended standard of Little [64]. Therefore, the research framework and conclusion of this study will present a good generalized validity.

### 4.3. Main Effect Analysis of the Structural Model

This study used the multi-group analysis method recommended by Chin [65] to examine the hypothesis of the moderating effect of country in the research model. The significance of the relationships in each subgroup was evaluated through calculating the path coefficients and t values of the hypothesized relationships. The standardized structural weights for the Taiwanese subgroup and mainland Chinese subgroup are shown in Figure 2; Figure 3, respectively. These standard structural weights were estimated with the item-factor loadings held equal across regions. As a result, they are proved to be the best estimates of the true structural weights. Figure 2 shows that all six hypotheses were supported for the Taiwanese subgroup, whereas only four hypotheses were supported for the Chinese subgroup. To sum up, H1 is not significant, H2 is partially significant, and H3, H4, H5 and H6 are supported.

Differences between Taiwanese and Chinese subgroups were found through the statistical analysis. The objective was then to determine the significance of the differences. First, the data were tested using the Kolmogorov–Smirnov test of normality, and the results indicated that the distribution was not normal. Therefore, the Smith–Satterthwaite t test, which is utilized when the data violate the normal distribution or for unequal variances, was selected [65], and the results of the t test for each subgroup are detailed in Table 4. There were significant differences in all path coefficients between the two subgroups. Other than the path coefficient of POS→ Job Security(JS), which was negatively significantly different, the remaining five path coefficients were positively significantly different, thus providing support for H7. The results of hypothesis testing are discussed further in the following sections for their possible implications for science teaching.

### 4.4. Mediating Role of Perceived Employability

In order to learn about the significance of perceived employability and job security in this study and further examine the mediating effect of perceived employability for samples A, B and total, respectively, this study adopted the Sobel test to examine the mediation model as previously proposed [66]. This test is designed to assess whether a mediating variable (perceived employability) carries the effects of the independent variables (Job security and Perceived organizational support) to a dependent variable (Job satisfaction). As Table 5 shows, results of six separate Sobel tests for the Taiwan sample and Total sample supported PE mediating the relationship between perceived employability (PE)/POS and JST. Also, the results support JS mediating the relationship between POS and JST. The Sobel tests were all significant in the Taiwan and Total samples, which means that the perceived employability and job security are exerting a significant mediating effect. This study shows the results on all hypotheses in Table 6, including for both Taiwan and mainland China.

## 5. Discussion

Taking the employees in the service industries as the research sample, in this paper, a competence enhancement model was used to examine the relationships among job security, perceived organizational support, perceived employability and job satisfaction, thus to bridge the theoretical gap of applying Western theory to the Asian situation, and to enhance the generalization level of the theory. In addition, this study also followed the suggestions of scholars to find the crux from the real world to verify the differences in the relationships among variables in the research framework of Taiwan and mainland China [67,68]. According to the results, the following findings can be concluded.

Firstly, the results of the study show that, for both the Taiwan and mainland China research samples, the perceived organizational support has a positive impact on job security, employability and job satisfaction. This means that when the organization or supervisors take the initiative to provide employees formal and informal assistance at work, the employees will feel the organizational identification and goodwill, which can not only improve their job security and job satisfaction, but also promote them to strengthen individual problem-solving ability at work and their perceived employability [1].

Furthermore, it has been found for all the paths of perceived organizational support that, for both the Taiwan and mainland China samples, the job security coefficient is the highest. This means that most employees in Asia have suffered from job insecurity, which can contribute to the employees’ responsibility of supporting the family (parents and children). Even so, job insecurity may have a negative impact on their work attitudes. When they obtain the affirmation and support from the organization, their sense of job insecurity will be lower.

In addition, the difference analysis of the path coefficient demonstrates the influence of the perceived organizational support on employability and job satisfaction. The perception of Taiwan samples is stronger than that for mainland China, indicating that Taiwan employees regard the support from the organization or supervisors as the power source of enhancing their employability. Only by continuously improving competitiveness can they obtain a high degree of sense of achievement and job satisfaction [1,11]. The interesting thing is, for mainland China samples, the influence of the perceived organizational support on the perception of job security is greater than that of Taiwan samples. Analysis of the political and economic development process of mainland China shows that that mainland China employees have strong recognition and confidence in their country and organization because of the ideology of socialism and “loving the factory as your family” [68]. In other words, as employees are highly trusted in the organization, the perceived organizational support will enhance their job security, as well as motivate employees to work hard, and hence, enhance their job performance.

Secondly, the study assumes that job security is negatively related to employability, but further analysis has proved that this hypothesis is false. For Taiwan samples, job security has a significant positive relationship with employability, which indicates that employees will continue to enhance their strengths and competence in a comfortable work environment. A possible reason for this situation is that Taiwan employment comes with less competitive pressure and lower salary levels, so that even if they have higher sense of job security, they will still work harder to get access to lucrative jobs. This finding implies that Taiwan employees will attribute their success or failure to their individual ability. They deem their success as the results of efforts and their failure as due to incompetence and insufficient preparation. Although the research result is contrary to that of Cuyper et al. [16], it is consistent with their findings, namely, job security is positively related to employability.

Thirdly, this study assumes that job security is positively related to job satisfaction. It has been found that job security has a significant positive relationship with job satisfaction in Taiwan samples but is not significant in mainland China samples, so this hypothesis is partially correct. The intriguing results showed that job security has an insignificant negative relationship with job satisfaction, which is inconsistent with the findings of Mak and Mueller [32], Strazdins et al. [33]. This situation may be caused by three factors. Firstly, the scale used in this study was developed by Western scholars, so interregional factors have led to different results. Secondly, employees’ sense of job security does not necessarily lead to positive job satisfaction. The organizational culture of China stresses authority, hierarchy and high-power distance, so even if employees have a stable working environment, it is still difficult for them to obtain the authority from the organization and their supervisors to deal with professional problems in the work. Based on Maslow’s hierarchy of needs theory, this study infers that the mainland employees call for emotional demand but have relatively lower demands for respect and self-actualization on the higher positions of the hierarchy.

Fourthly, this study assumes that perceived employability has a positive relationship with job satisfaction. It has been confirmed in this study that the perceived employability has a positive relationship with job satisfaction in both sample groups, and this result is consistent with the findings of Gamboa, Gracia, Ripoll, and Peiró [69], namely, that improvement of employees’ perceived employability will promote their job satisfaction. In addition, the difference analysis on the path coefficient of the two groups shows that the influence of perceived employability on job satisfaction is greater in Taiwan than mainland China. As Fugate et al. [10] note, individual employability is the integration of human capital, social capital, professional identity and personality adjustment. Similarly, Taiwan employees solve a variety of professional problems in daily work through organizational authorization and support, and thus cultivate personal human capital to meet employers’ expectations, to achieve individual career goals, and to obtain higher job satisfaction.

### 5.1. Practical Implications

The study also has significant practical implications. Firstly, it is found in this study that in both Taiwan and mainland China, employees’ perceived organizational support has a positive impact on job security, job satisfaction and perceived employability. This means that perceived organizational support, no matter in the Asian or Western contexts, is a significant positive physiological and psychological factor affecting employees. Both perceived organizational support and job satisfaction can be regarded as a predictor of job performance. Therefore, this study suggests that organizational management can adopt the practical methods that have been confirmed by scholars to enhance employees’ perceived organizational support, such as enhancing perceived organizational justice [70] and employees’ job autonomy [41], so as to enhance employees’ sense of organizational support.

Secondly, the core dimension of this study is employees’ perceived employability. The research result has shown that perceived organizational support has a positive impact on employability. From the perspective of organizational ability, employees solve their problems through organizational support, and thus enhance personal human capital and their perceived employability. In return, organizational ability can be enhanced as well. Although some scholars believe that educational training and on-the-job learning can enhance employees’ employability, it still cannot succeed without the support from colleagues, supervisors and the organization to create a motivating learning environment [1], and thus achieve a good organization performance and working environment.

Thirdly, in terms of the insignificant part of mainland samples in the research framework, this study can provide practical suggestions on human resource management acquired from Taiwan businessmen who invest in mainland China. Unlike Taiwan employees, mainland employees do not have to maintain strong learning initiative after entering the job market. As a result, in this study, it is suggested that Taiwan management, first of all, should strengthen the employees’ learning motivation, create a competitive working environment and enhance employees’ job autonomy through authorization to improve employees’ sense of job security, so as to enhance their perceived employability, job satisfaction and job performance.

### 5.2. Limitations and Suggestions for Future Research

The research data of this study was acquired from a 2015 questionnaire, which reflects only a single point in time rather than completely presenting the dynamic process of employees’ job insecurity. This also neglects the influence of time and social changes on the research results. Reviewing the past related research on job insecurity, in addition to the longitudinal study of Dekker and Schaufeli [38], Mauno and Kinnunen [71] and Hellgren and Sverke [72], all other studies are cross-sectional. Therefore, in this study, it is suggested that more longitudinal research is conducted into job insecurity in the future, to further explore the influence of job insecurity on employees and the organization.

Moreover, in this study, questionnaires developed by different scholars were adopted to process data collection and analysis, so that it is not clear whether there are other factors affecting employability. Therefore, focus group interviews and other qualitative data collection methods can be used to help deepen the analysis of the research results, the understandings of the current situation, and the difficulties of the organization in promoting employability. Therefore, it is suggested, in this study, that different independent variables or related situational variables can be added to further research to enhance employees’ perceived employability.

Purposive sampling, instead of random sampling, was utilized in this study. Although the samples of mainland China were collected from many provinces, only enterprises in Zhejiang and Fujian provinces were subject to the questionnaire survey due to the limitations in time and funds. The difference in GDP of each province of mainland China may result in the diversity of research results. Thus, future researchers are suggested to expand the scope of investigation, find the distinction between provinces before the statistical analysis, and then improve the degree of generalization of research results. 

Finally, in the complex work environment, in addition to employees, other stakeholders include supervisors and colleagues. However, this study mainly discusses employees’ perception of employability, and does not investigate whether knowledge and skills have met the expectations of management and the organization. Therefore, it is suggested that dual level data be adopted in further measurements of employability as a comparison, and the influence of cross-level factors can be discussed in different groups, thereby helping understand the cognitive differences of employees and other stakeholders in the work environment in terms of perceived employability.

## Figures and Tables

**Figure 1 ijerph-16-02613-f001:**
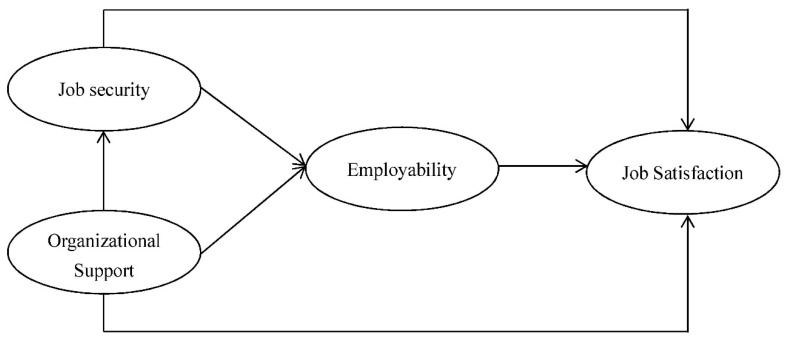
Research framework.

**Figure 2 ijerph-16-02613-f002:**
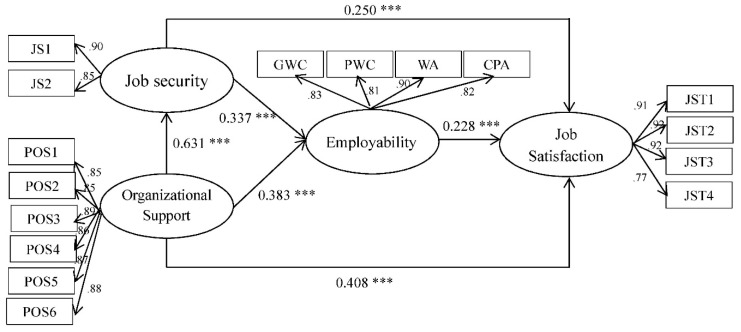
Structural model on Taiwanese sample. Note: **** p* < 0.001. JST means “Job satisfaction”, JS means “Job Security”

**Figure 3 ijerph-16-02613-f003:**
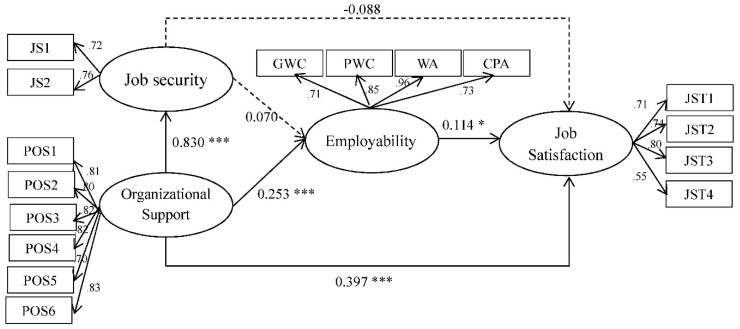
Structural model on Chinese sample. Note: * *p* < 0.05, **** p* < 0.001.

**Table 1 ijerph-16-02613-t001:** Sample.

Variables	Indicators	Taiwan	China
Gender	Male	360	333
	Female	399	215
Age	20–30	499	274
	31–40	144	158
	41–50	95	90
	Above 51	21	26
Work experience	1–3 years	467	254
	3–5 years	119	141
	5–7 years	121	85
	Above 7 years	52	68

**Table 2 ijerph-16-02613-t002:** Scales.

Constructs	Variables	Scales
Employability	General ability for work	Expression and communication
		Time management
		Leadership
		Innovation
		Team work
		Native language
		Foreign language
		Stability and pressure resistance
	Professional ability for work	Professional knowledge and skill
		Computer literacy
		Application of theory to work
		Problem finding and solving
	Attitude at work	Learning desire
		Plasticity
		Understanding of professional ethics
	Career planning and confidence	Understanding and planning of individual career development
		Understanding of environment and development of industries
		Job search and self-promotion
Job insecurity	Job insecurity	I can keep my current job for as long as I want it.
		This job has retirement security.
		I can be sure of my present job as long as I do good work.
		I am not really sure how long my present job will last.
		I am afraid of losing my present job.
Perceived organizational support	Perceived organizational support	I get the necessary time and means to further develop my competencies.
		I can make use of a personal development plan to know what competencies I need to develop and how I can develop them best.
		My boss regularly gives me feedback about my performance.
		My organization provides new and creative training opportunities.
		My boss makes sure that I can learn on the job by giving me challenging assignments.
		My colleagues regularly give me feedback about my performance.
		I can regularly change jobs within my company (without promotion) to develop new competencies.
		My boss makes sure that I develop the competencies that I need for my career.
		All information about career opportunities in the organization is readily available.
		I have been given tasks that develop my competencies for the future.
		I have been given a personal development plan to better understand my possibilities within the organization and the competencies I need to fully exploit them.
		I have been given the possibility within my organization to develop the competencies I need to get a promotion and move to a function at a higher level of the organization.
Job satisfaction	Job satisfaction	How satisfied are you with your work performance?
		How satisfied are you with the quality of your work performance?
		How satisfied are you with the way you perform your work?
		How satisfied or dissatisfied are you with the way you carry out your work activities?
		How satisfied are you with your collaboration with your supervisor?
		How satisfied are you with the support you get from your supervisor?
		How satisfied are you with the support you give to your supervisor?

**Table 3 ijerph-16-02613-t003:** Measurement.

		1	2	3	4	5	6	7
1. GAW		0.74/0.65	0.143 **	0.159 **	0.118 **	0.134 **	0.112 **	0.146 **
2. PAW		0.811 **	0.79/0.77	0.816 **	0.623 **	0.155 **	0.189 **	0.153 **
3. AW		0.752 **	0.749 **	0.77/0.72	0.699 **	0.237 **	0.289 **	0.185 **
4. CPC		0.677 **	0.659 **	0.739 **	0.83/0.84	0.228 **	0.297 **	0.181 **
5. Job Security		0.459 **	0.389 **	0.423 **	0.451 **	/0.74	0.640 **	0.199 **
6. POS		0.483 **	0.429 **	0.512 **	0.528 **	0.543 **	0.87/0.71	0.307 **
7. Job Satisfaction		0.510 **	0.470 **	0.543 **	0.506 **	0.536 **	0.668 **	0.88/0.81
Mean	Taiwan	3.56	3.68	3.67	3.57	3.42	3.54	3.68
	China	3.70	3.87	3.91	4.00	4.08	4.31	3.85
SD	Taiwan	0.65	0.70	0.72	0.74	0.62	0.67	0.64
	China	0.63	0.72	0.69	0.71	0.50	0.48	0.39
Cronbach’s α	Taiwan	0.902	0.871	0.805	0.863	0.550	0.947	0.919
	China	0.746	0.853	0.761	0.879	0.823	0.913	0.772
AVE	Taiwan	0.547	0.630	0.599	0.680	0.544	0.756	0.782
	China	0.416	0.595	0.524	0.710	0.543	0.498	0.663
CR	Taiwan	0.905	0.871	0.817	0.864	0.870	0.949	0.934
	China	0.809	0.855	0.767	0.880	0.703	0.795	0.921

Note: GAW means “General ability for work”, PAW means “Professional ability for work”, AW means “Attitude at work”, CPC means “Career planning and confidence”, POS means “Perceived organizational support”. ** *p* < 0.01.

**Table 4 ijerph-16-02613-t004:** Multi-group testing.

Model	*χ* ^2^	*df*	*χ* ^2^ */df*	*p*	RMSEA	NFI	ECVI	90% CI
1. Unconstrained	479.80	184	2.608	0.000	0.035	0.971	0.503	(0.456–0.555)
2. Measurement weights	718.38	196	3.665	0.000	0.045	0.956	0.667	(0.607–0.733)
3. Structural weights	821.60	202	4.067	0.000	0.048	0.950	0.737	(0.672–0.807)
4. Structural covariances	904.29	203	4.455	0.000	0.051	0.944	0.799	(0.730–0.873)
5. Structural residuals	1097.20	206	5.326	0.000	0.058	0.933	0.942	(0.865–1.024)
6. Measurement residuals	1429.27	222	6.438	0.000	0.065	0.912	1.172	(1.083–1.266)
2-1	238.58	12		0.000		0.015		
3-1	341.80	18		0.000		0.021		
4-1	424.49	19		0.000		0.026		
5-1	617.41	22		0.000		0.038		
6-1	949.48	38		0.000		0.058		

**Table 5 ijerph-16-02613-t005:** Testing of mediating effect.

Mediating Path	Individual Path	Taiwan		Mainland China		Total	
		Path Coefficient	Sobel Test	Path Coefficient	Sobel Test	Path Coefficient	Sobel Test
JS→PE→JST	JS→PE	0.337 (0.040)	4.887 ***	0.070 (0.153)	0.454	0.286 (0.026)	4.919 ***
	PE→JST	0.228 (0.038)		0.114 (0.031)		0.231 (0.042)	
POS→PE→JST	POS→PE	0.383 (0.042)	5.012 ***	0.253 (0.145)	1.576 ^†^	0.306 (0.026)	4.983 ***
	PE→JST	0.228 (0.038)		0.114 (0.031)		0.231 (0.042)	
POS→JS→JST	POS→JS	0.631 (0.040)	6.507 ***	0.830 (0.058)	−0.870	0.745 (0.027)	6.384 ***
	JS→JST	0.250 (0.035)		−0.088 (0.101)		0.210 (0.032)	

Note: ^†^
*p* < 0.1, * *p* < 0.05, ** *p* < 0.01, **** p* < 0.001. → means path

**Table 6 ijerph-16-02613-t006:** Results of hypotheses testing.

Hypotheses	Taiwan		Mainland China	
	Coefficients	Results	Coefficients	Results
H1 JS→PE	0.337 ***	Support	0.007	Not support
H2 JS→JST	0.250 ***	Support	−0.088	Not support
H3 POS→JST	0.631 ***	Support	0.830 ***	Support
H4 POS→PE	0.383 ***	Support	0.253 ***	Support
H5 POS→JS	0.408 ***	Support	0.397 ***	Support
H6 PE→JST	0.228 ***	Support	0.114 *	Support

Note: * *p* < 0.05, **** p* < 0.001.

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
