# Peer review of "A Comparative Study of the Relationship among Antecedents and Job Satisfaction in Taiwan and Mainland China: Employability as Mediator"

_ijerph, 2019, doi:10.3390/ijerph16142613_

Round 1
Reviewer 1 Report
This is a well-designed and a well-done research, the results are convinced. I have the following suggestions:
1. In the section of conclusion, not in each case (response to the previous hypotheses) Taiwan and Mainland China are both mentioned, for instance, for the hypothesis "job security is negatively related to employability" only Taiwan has been mentioned, not Mainland China. I suggest applying a tabular display to show the results on all hypotheses, including both Taiwan and Mainland. It will then help to present the results clearer.
2. I suggest using the interregional comparison instead of cross-country or cross-culture comparison between Taiwan and Mainland China. However, that is only my own interpretation.
3. In the section on method the author/s shall explain the comparability between Taiwan and Mainland China, why not consider undertaking a comparison between Taiwan and other Chinese provinces, why are Mainland China and Taiwan comparable at all?
Author Response
Reviewer’s Comment 1:
1. In the section of conclusion, not in each case (response to the previous hypotheses) Taiwan and Mainland China are both mentioned, for instance, for the hypothesis "job security is negatively related to employability" only Taiwan has been mentioned, not Mainland China. I suggest applying a tabular display to show the results on all hypotheses, including both Taiwan and Mainland. It will then help to present the results clearer.
Author’s response:
Many thanks for the reviewer’s comment. We have added a tabular display to show the results on all hypotheses in Table 6, including both Taiwan and Mainland China.
Table 6. Results of Hypotheses Testing
Hypotheses | Taiwan | Mainland China | ||
Coefficients | Results | Coefficients | Results | |
H1 JS→PE | 0.337*** | Support | 0.007 | Not support |
H2 JS→JST | 0.250*** | Support | -0.088 | Not support |
H3 POS→JST | 0.631*** | Support | 0.830*** | Support |
H4 POS→PE | 0.383*** | Support | 0.253*** | Support |
H5 POS→JS | 0.408*** | Support | 0.397*** | Support |
H6 PE→JST | 0.228*** | Support | 0.114* | Support |
Note: * p < .05, *** p < .001
[line 446-447]
Reviewer’s Comment:
2. I suggest using the interregional comparison instead of cross-country or cross-culture comparison between Taiwan and Mainland China. However, that is only my own interpretation.
Author’s response:
Many thanks for the reviewer’s comment. We have used the interregional comparison instead of cross-country or cross-culture comparison between Taiwan and Mainland China.
[On the other hand, scholars believe that the interregional comparison results can provide significant insights, so that most scholars adopted interregional ways to compare the different research results in different contexts] (line 63-65)
Reviewer’s Comment:
3. In the section on method the author/s shall explain the comparability between Taiwan and Mainland China, why not consider undertaking a comparison between Taiwan and other Chinese provinces, why are Mainland China and Taiwan comparable at all?
Author’s response:
Many thanks for the reviewer’s comment. This study uses the purposive sampling to compare Taiwan and mainland China due to the initial research background and motivations. But we agree with the opinions of the reviewer, because provinces in mainland China develop at different levels and have differences in GDP, which may have a certain effect on this study. Most samples of this study were collected from Fujian and Zhejiang provinces, which are two coastal provinces in mainland China. Therefore, this study will add the source of these samples in 3.1 Sampling, and suggests future studies to expand the sampling scope and enhance the degree of generalization in the research limitations.
[and most questionnaires were collected from Fujian and Zhejiang provinces.] (line 284-285)
[The purposive sampling, instead of the random sampling, was utilised in this study. Although the samples of mainland China were collected from many provinces, only enterprises in Zhejiang and Fujian provinces were subject to the questionnaire survey due to the limitations in time and funds. The difference in GDP of each province of mainland China may result in the diversity of research results. Thus, future researchers are suggested to expand the scope of investigation, find the distinction between provinces before the statistical analysis, and then improve the degree of generalization of research results.] (line 568-574)

Reviewer 2 Report
I have appreciated very much your work and the effort in conducting the empirical study.
I have not substantial revisions to require, but only some suggestions to improve the quality of the manuscript.
The authors stated that: "Although job satisfaction can be regarded as the predictor of job performance, few empirical researches have been made to explore the relationship between employees’ employability and job satisfaction (De Vos et al., 2011; Ng, Eby, Sorensen, and Feldman, 2005)". However a very recent research related employability with life satisfaction and quality of life; it could be cited:
Magnano, P., Santisi, G., Zammitti, A., Zarbo, R., & Di Nuovo, S. (2019). Self-Perceived Employability and Meaningful Work: The Mediating Role of Courage on Quality of Life. Sustainability (Switzerland), 11(3), 764–778. https://doi.org/10.3390/SU11030764
p. 7, line 280-281: the authors wrote: "557 copies of questionnaires have been collected and 548 copies are invalid": maybe they want to say that 548 were valid, considering the subsequent table presenting the sample
It is not clear if the scale selected to measure the variables have been used in the original form or they have been modified by the authors (in this second case, it should be clarified why and how they have been modified). In my opinion, the measures should be presented in a more clear way.
Author Response
Reviewer’s Comment 1:
The authors stated that: "Although job satisfaction can be regarded as the predictor of job performance, few empirical researches have been made to explore the relationship between employees’ employability and job satisfaction (De Vos et al., 2011; Ng, Eby, Sorensen, and Feldman, 2005)". However a very recent research related employability with life satisfaction and quality of life; it could be cited:
Magnano, P., Santisi, G., Zammitti, A., Zarbo, R., & Di Nuovo, S. (2019). Self-Perceived Employability and Meaningful Work: The Mediating Role of Courage on Quality of Life. Sustainability (Switzerland), 11(3), 764–778. https://doi.org/10.3390/SU11030764
Author’s response:
Many thanks for the reviewer’s comment. I added this reference into the revision and made our sentences more meaningful.
Reviewer’s Comment 2:
p. 7, line 280-281: the authors wrote: "557 copies of questionnaires have been collected and 548 copies are invalid": maybe they want to say that 548 were valid, considering the subsequent table presenting the sample
Author’s response:
Many thanks for the reviewer’s comment. This is a typo, which has been revised in this manuscript.
Reviewer’s Comment 3:
It is not clear if the scale selected to measure the variables have been used in the original form or they have been modified by the authors (in this second case, it should be clarified why and how they have been modified). In my opinion, the measures should be presented in a more clear way.
Author’s response:
Many thanks for the reviewer’s comment. We added a table to present our measure of variable more clear.
Table 2. Scales
Constructs | Variables | Scales |
Employability | General ability for work | Expression and communication Time management Leadership Innovation Team work Native language Foreign language Stability and pressure resistance |
Professional ability for work | Professional knowledge and skill Computer literacy Application of theory to work Problem finding and solving | |
Attitude at work | Learning desire Plasticity Understanding of professional ethics | |
Career planning and confidence | Understanding and planning of individual career development Understanding of environment and development of industries Job search and self-promotion | |
Job insecurity | Job insecurity | I can keep my current job for as long as I want it. This job has retirement security. I can be sure of my present job as long as I do good work. I am not really sure how long my present job will last. I am afraid of losing my present job. |
Perceived organizational support | Perceived organizational support | I get the necessary time and means to further develop my competencies. I can make use of a personal development plan to know what competencies I need to develop and how I can develop them best. My boss regularly gives me feedback about my performance. My organization provides new and creative training opportunities. My boss makes sure that I can learn on the job by giving me challenging assignments. My colleagues regularly give me feedback about my performance. I can regularly change jobs within my company (without promotion) to develop new competencies. My boss makes sure that I develop the competencies that I need for my career. All information about career opportunities in the organization is readily available. I have been given tasks that develop my competencies for the future. I have been given a personal development plan to better understand my possibilities within the organization and the competencies I need to fully exploit them. I have been given the possibility within my organization to develop the competencies I need to get a promotion and move to a function at a higher level of the organization. |
Job satisfaction | Job satisfaction | How satisfied are you with your work performance. How satisfied are you with the quality of your work performance. How satisfied are you with the way you perform your work. How satisfied or dissatisfied are you with the way you carry out your work activities. How satisfied are you with your collaboration with your supervisor. How satisfied are you with the support you get from your supervisor. How satisfied are you with the support you give to your supervisor. |
(line 311)
